# The Promoting Role of HK II in Tumor Development and the Research Progress of Its Inhibitors

**DOI:** 10.3390/molecules29010075

**Published:** 2023-12-22

**Authors:** Bingru Liu, Yu Lu, Ayijiang Taledaohan, Shi Qiao, Qingyan Li, Yuji Wang

**Affiliations:** 1Department of Medicinal Chemistry, College of Pharmaceutical Sciences of Capital Medical University, Beijing 100069, China; lbr9904@163.com (B.L.); ly_950420@163.com (Y.L.); ayijiang227@ccmu.edu.cn (A.T.); 2Beijing Area Major Laboratory of Peptide and Small Molecular Drugs, Engineering Research Center of Endogenous Prophylactic of Ministry of Education of China, Beijing Laboratory of Biomedical Materials, Laboratory for Clinical Medicine, Capital Medical University, Beijing 100069, China; 3Department of Core Facility Center, Capital Medical University, Beijing 100069, China; 4Civil Aviation Medical Center, Civil Aviation Administration of China, Beijing 100123, China; shiqiao170717@163.com

**Keywords:** hexokinase II, antitumor, Warburg effect, glycolysis, inhibitors

## Abstract

Increased glycolysis is a key characteristic of malignant cells that contributes to their high proliferation rates and ability to develop drug resistance. The glycolysis rate-limiting enzyme hexokinase II (HK II) is overexpressed in most tumor cells and significantly affects tumor development. This paper examines the structure of HK II and the specific biological factors that influence its role in tumor development, as well as the potential of HK II inhibitors in antitumor therapy. Furthermore, we identify and discuss the inhibitors of HK II that have been reported in the literature.

## 1. Introduction

In 2022, the National Cancer Center published data on the incidence and mortality rates of cancer in China in 2016. The report revealed that there were approximately 4,064,000 newly diagnosed cancer cases and 2,413,500 cancer-related deaths that year [1]. Furthermore, researchers project that the number of cancer cases will continue to rise in the coming decade. The analysis conducted by Rebecca L. Siegel, MPH, and colleagues focused on cancer incidence and mortality trends in the United States since 1973. Although the overall cancer mortality rates in the United States have progressively declined by 33%, the incidences of certain types of cancer are still increasing [2]. The intricate physiological processes of cancer cells, the development of drug resistance, and the adverse effects that come with chemotherapy dosage make the management of cancer a profoundly complex and formidable global concern. The ability of cancer cells to reprogram their metabolisms renders them resistant to singular drug interventions aimed at modulating a solitary target. As a result, the pursuit of enhanced therapeutic efficacy in cancer chemotherapy has led to a burgeoning interest in the concept of drug synergism, wherein the simultaneous targeting of two or more cancer progression-related biomarkers has exhibited promising outcomes [3,4,5].

Unlike normal tissues, tumor cells exhibit a distinct metabolic behavior known as “aerobic glycolysis” or the Warburg effect, in which they preferentially metabolize glucose to lactate, even in the presence of ample oxygen that could support mitochondrial oxidative phosphorylation. This aberrant metabolic state hampers the ability of cancer cells to switch back to normal energy production pathways, leading to a sustained reliance on glycolysis and consequent inability to regain their typical physiology [6]. In 2009, a comprehensive analysis by Lewis C. Cantley and Craig B. Thompson [7] scrutinized the “Warburg effect” and its impact on cancer cell metabolism and growth regulation. Within this framework, they posited that the metabolic behavior of cancer cells represents a crucial adaptation enabling all proliferating cells to optimize the assimilation and utilization of vital nutrients, such as nucleotides, amino acids, and lipids, ultimately culminating in cellular proliferation.

The initial step in glycolysis involves the conversion of glucose to glucose 6-phosphate (G-6-P) through a process of phosphorylation. A key enzyme that strongly influences the rate of this reaction is hexokinase II (HK II), which has consistently been found to be overexpressed in various types of cancer. Notably, inhibiting or reducing the expression of HK II has been shown to impede cancer cell proliferation in animal models while resulting in minimal adverse effects [8,9]. Moreover, Ruiqi Li [10] employed the Tumor Genome Atlas (TCGA) dataset and conducted a comprehensive gene expression analysis in 33 different tumors to investigate the potential implications of HK II. This investigation revealed a compelling association between HK II expression and the progression of particular tumor types, underscoring the viability of targeting HK II as a potential strategy for cancer therapy.

In this review, we present an in-depth analysis of the intricate structure of HK II, with particular emphasis on its distinctive N-terminus and the consequential catalytic activity facilitated by its unique linker. In order to gain a more profound comprehension of HK II’s intricate mechanism of action in vivo, we summarize the intricate pathways associated with HK II expression, encompassing crucial proteins, RNAs, and other related factors. Exploring the links between these entities may foster a more systematic approach in the development of novel pharmaceutical interventions aimed at manipulating HK II’s mechanism of action. Moreover, this review elucidates the interplay between HK II and cellular drug resistance and also highlights recent discoveries pertaining to HK II inhibitors in ongoing studies, where high-throughput screening and natural products occupy a primary position as potential sources. At present, no commercially available HK II inhibitors exist. Thus, our review aims to offer a practical comprehension of HK II’s mode of action within tumor cells, with the ultimate goal of facilitating the design of robust and highly selective HK II inhibitors.

## 2. Structural Features and Functional Implications of Hexokinase II

The hexokinase family is composed of four isoforms: hexokinases I, II, III, IV, and hexokinase domain containing 1 (HKDC1), that each have specific distributions within the body. Hexokinase I (HK I) is primarily found in the brain, HK II in skeletal muscle, HK III in leukocytes, and HK IV in the liver. Notably, both hexokinase I and II have been discovered to bind to mitochondria through an interaction with voltage-dependent anion channels (VDACs). This binding mechanism facilitates direct access of hexokinase to ATP generated by the mitochondria [11].

By using the UniProt database, researchers have determined that human HK II consists of 917 amino acids and has a mass of 102,380 Da and a size of 2.45 Å. Its crystal structure reveals a homodimeric arrangement, with two structurally identical N- and C-termini of equal size, and each half also comprises large and small domains (Figure 1). Moreover, the region that connects HK II’s two hexokinase structural domains plays a crucial role in facilitating the catalysis of the amino-terminal hexokinase structural domain. Additionally, the N-terminus serves as a regulator that influences the overall stability of the enzyme. This distinction underscores the significance of the structural domains in mediating the catalytic function and stability of HK II. These termini adopt an *α*/β fold and are connected via a lengthy eight-turn linker helix known as *α*13. Furthermore, helix α5, which is perpendicular to the linker helix *α*13, serves as a carrier for the mitochondrial ATP [12]. In particular, the *α*13 accommodates the N-terminal catalytic residue D657(C-D209) [12].

HK II is positioned in the outer mitochondrial membrane, where it acts as the primary rate-limiting enzyme responsible for facilitating the glycolysis of glucose. Importantly, it associates with VDAC1, a protein present on the outer membrane of mitochondria (MOM). By means of computer simulations, Dawei Zhang et al. [13] have hypothesized that the binding of HK II restricts the loop connecting the N-terminal α-helix to the barrel wall within VDAC1. This loop governs the helical movement through the channel pore and inhibits the transition of VDAC1 into a closed state in cancer cells.

Consequently, ADP and Pi enter the mitochondria through the c-state end of the AAC protein, participating in the TCA cycle. The ATP produced by the mitochondria is transported to the mitochondrial intermembrane space through the m-state end of the AAC protein, and then reaches HK through VDAC1, where it furnishes the necessary energy for HK II to catalyze the conversion of glucose into glucose-6-phosphate (G6P). The resulting G6P is utilized in subsequent glycolytic reactions to continuously generate ATP, thereby providing energy for crucial cellular functions. The lactate produced is then excreted out of the cell through transmembrane proteins (Figure 2) [14,15].

## 3. The Significance of HK II in Tumor Development and Progression

Several studies have highlighted the overexpression of HK II in the majority of tumor cells, along with its interdependence on various factors within the human biological system. A deeper understanding of the intricate associations between these biological elements can potentially elucidate the mechanisms underlying cancer cells’ unique metabolism. This deeper understanding may hold great promise for the development of better cancer drugs. Hence, the subsequent sections explore the impacts of proteins and nucleic acids on HK II expression, encompassing its up-regulation, down-regulation, and overall stability. In this section, we focus on the biological factors that influence the up-regulation (Figure 3) and down-regulation (Figure 4) of HK II, as well as the non-metabolic impact of HK II on tumor development.

### 3.1. Upward Adjustment of HK Pathways

#### 3.1.1. Proteins and Transcription Factors

Tyrosine kinase receptor 2 (ErbB2) is a receptor protein encoded by the proto-oncogene ErbB2. High expression of ErbB2 in cancer cells has been associated with enhanced cell proliferation and migration [16]. Notably, in breast cancer cells that overexpress ErbB2, recent research has revealed a dual role for tyrosine kinase receptor 2. First, it promotes the localization of HK II at the outer mitochondrial membrane. Second, it renders cancer cells more susceptible to glucose starvation and treatment with 3-bromopyruvic acid (3-BrPA) [17]. This newly discovered in vitro and in vivo mechanism of ErbB2-activated glycolysis in breast cancer cells has contributed significantly to the advancement of precision therapeutic approaches.

HectH9, an oncogene-encoded protein, encompasses a c-terminal hect structural domain that functions as an e3 ubiquitin ligase. This ligase is involved in the regulatory mechanisms of several pivotal proteins, including p53 and Myc, notwithstanding conflicting reports pertaining to its influence on tumor development [18,19,20]. Importantly, HectH9 has been shown to modulate the k63 ligation of HK II within the glycolytic pathway by facilitating ubiquitination. This in turn regulates the subcellular localization of HK II within the mitochondria, ultimately promoting glycolysis and preventing apoptosis. These findings implicate HectH9 in tumor promotion [21]. Conversely, a distinct protein called TRIM59 has been observed in non-small-cell lung cancer (NSCLC) cells to enhance glycolytic activity by inducing PTEN ubiquitination. This process amplifies AKT activation, which subsequently up-regulates HK II expression [22]. Some researchers have proposed that HectH9 may have a context-dependent role in tumor pathogenesis. However, the intricate and multifaceted nature of biological functions, encompassing aspects such as cellular proliferation, differentiation, signaling pathways, and responsiveness to environmental stimuli, present significant challenges to arriving at a unified consensus.

BACH1, a transcription factor, plays a crucial role both in tumor metastasis and tumor metabolism. Inhibition of BACH1 has been shown to suppress cancer cell metastasis and proliferation, while also enhancing sensitivity to mitochondrial inhibitors [23,24]. Furthermore, activation of Nrf2 leads to a significant increase in BACH1 activity, irrespective of the presence of exogenous antioxidants. This in turn promotes the transcription of HK II, thereby boosting glucose uptake and facilitating cancer cell metastasis [25]. Given that tumors characterized by high BACH1 levels often arise in patients with NRF2/KEAP1 mutations, targeting BACH1 or its upstream/downstream proteins represents a promising strategy to combat lung cancer metastasis.

Additionally, the phosphorylation of HK II Thr473 by the mouse pre-B lymphoma virus insert site 2 (PIM2) promotes glycolysis and confers resistance to paclitaxel in breast cancer cells [26], and here the consumption rate of substrates and the production of enzymes exhibit a proportional relationship. In the context of colorectal cancer (CRC) cells, overexpression of the immune regulatory protein B7-H3 has also been observed to augment substrate consumption rates, thereby favoring tumor growth [27]. The signal transducer and activator of transcription 3 (STAT3) regulates glucose metabolism in hepatocellular carcinoma (HCC) through the HK II mTOR-mediated pathway, contributing to the development of HCC [28]. Calcium/calmodulin-dependent protein kinase kinase β (CaMKKβ) is a multifunctional protein kinase. Overexpression studies have found that CaMKKβ is overexpressed in many types of cancer. In HCC, CaMKKβ inhibits cell proliferation, invasion, and glycolysis through the PI3K/AKT pathway, and increases cell apoptosis [29].

#### 3.1.2. Signaling Pathways

The role of various signaling pathways in HK II expression has been extensively investigated in academic research. Among these pathways, the Wnt/β-catenin signaling pathway has garnered significant attention due to its involvement in a multitude of critical functions, including the regulation of obesity and stem cell activity [30]. A key component of this pathway is β-catenin, which helps to regulate tumor development by binding to Wnt and translocating into the cell nucleus to modulate the expression of target genes [31,32]. Activation of the Wnt/β-catenin signaling pathway has also been found to promote the growth of diverse tumors, such as NSCLC, CRC, and gastric cancer [33,34,35,36]. Notably, in human epithelial ovarian cancer cells, HK II has been shown to facilitate proliferation and tumor formation by up-regulating Cyclin D1/c-myc through the Wnt/β-catenin pathway [37]. Overall, these findings highlight the critical role of the Wnt/β-catenin signaling pathway in mediating HK II expression and its implications in cancer biology. 

Upstream cytokines can likewise exert regulatory effects on HK II through various mechanisms. Acidic fibroblast growth factor (aFGF) has garnered significant attention in the field of diabetic vascular complication treatment due to its potent antioxidant properties. Recent research has unveiled that the expression of HK II is augmented, and its mitochondrial localization enhanced, through the activation of the Wnt/β-catenin/c-Myc axis mediated by aFGF [38]. Conversely, HK II can also govern cellular development by modulating other cytokines. For instance, it up-regulates cyclin A1 and down-regulates p27 expression via the Raf/MEK/ERK signaling pathway, thereby promoting the proliferation of cervical cancer cells [39]. This intricate interplay between HK II, upstream cytokines, and regulatory pathways highlights the multifaceted nature of cellular processes and underscores the need for further exploration in this domain. 

For example, tumor cells employ immune evasion by up-regulating PD-L1 expression, and the epidermal growth factor receptor (EGFR)-p38 mitogen-activated protein kinase (MAPK) pathway enhances PD-L1 levels via miR-675-5p and concurrently represses human leukocyte antigen-ABC (HLA-ABC) through HK II. This novel finding highlights a promising avenue for advancing cellular immunotherapy within the context of hepatocellular carcinoma [40].

Non-coding RNAs (ncRNAs) are a class of functional RNA molecules that cannot be further translated into proteins. Numerous studies have indicated their significant impact on glucose metabolism, directly targeting glycolytic enzymes, and indirectly affecting oncogenic signaling pathways [41,42]. In the case of 5-fluorouracil-resistant CRC cells, the regulatory role of FGD5-AS1 in glycolysis promotion has been observed with respect to the miR-330-3p-HK II axis. Consequently, targeting FGD5-AS1 has the potential to enhance the antitumor activity of 5-fluorouracil [43]. Another noteworthy ncRNA is cyclic RNA (circRNA) derived from atypical splicing of a gene’s pre-mRNA, that terminally possesses covalently linked RNAs. This unique characteristic enables it to influence tumor progression by modulating diffusion [44,45,46,47]. In triple-negative breast cancer (TNBC) cells, high circWHSC1 expression is associated with a poorer prognosis in TNBC patients. Here, circWHSC1 functions as a sponge for miR-212-5p, and the inhibition of circWHSC1 or elevation of miR-212-5p levels can effectively suppress glycolysis and impede TNBC progression. Furthermore, researchers have found that miR-212-5p directly targets AKT3 and that this overexpression of AKT3 counteracts the inhibitory effects of miR-212-5p [48].

In addition, knockdown of the hsa_circ_0069094 gene leads to the sequestration of miR-661, which subsequently regulates the expression of HMGA1. This regulatory mechanism influences breast cancer cell carcinogenesis and cellular glycolysis in breast cancer (BC) cells [49]. Moreover, circRHOBTB3 negatively modulates the PI3K/AKT signaling pathway, resulting in the inhibition of cell proliferation, metastasis, and glycolysis in ovarian cancer [50]. However, circ_0102273 acts as a sponge for miR-1236-3p, thereby regulating the expression of PFKFB3 [51]. This regulatory mechanism also contributes to the inhibition of cell proliferation, metastasis, and glycolysis and functions as a suppressant in ovarian cancer. Overall, these findings underscore the critical roles of circRNAs in regulating cellular processes such as glycolysis, TNBC progression, breast cancer cell carcinogenesis, and ovarian cancer metastasis.

The aforementioned studies primarily focus on the final manifestation of HK II, without delving into the specific stage at which its activity occurs. Conversely, other studies have shed light on the regulatory mechanism that governs HK II. For example, the pivotal role of hypoxia-inducible factor-1α (HIF-1α) in tumor development has been identified, where HIF-1α triggers the expression of HK II by binding to its promoter. Further investigation has revealed that miR-487a binds to the 3′-UTR of HIF-1α, thereby suppressing its expression. Interestingly, circRNF20 functions as an miRNA sponge, harboring miR-487a and counteracting its inhibitory effect. As a result, elevated levels of circRNF20 mitigate the suppression of miR-487a and ultimately foster the progression of breast cancer (BC). Moreover, IL-1β, known to enhance HIF-1α expression, similarly amplifies HK II expression [52]. 

Silent regulator protein 6 (SIRT6) and myeloid zinc finger protein (MZF1) function as negative regulators of HK II. Their complex, SIRT6-MZF1, acts as a repressor of HK II gene transcription by recruiting to the MZF1 site on the HK II promoter. Conversely, down-regulation of HIF-1α rescues the alterations in the expression of glycolysis genes that are dependent on SIRT6 [53,54]. HIF-1α also plays a significant role in glycolysis. Furthermore, miR-455 inhibits the mRNA expression of HK II by directly interacting with its mRNA, functioning as a tumor suppressor, and the non-coding RNA DLEU2 induces the expression of HK II by competitively binding to miR-455 [55]. miR-181b modulates the expression of HIF-1α by directly targeting the binding site within the 3′-untranslated region (3′-UTR), thereby concomitantly regulating HK II. Additionally, the involvement of enhancer zeste homolog 2 (EZH2) in the metabolic process of prostate cancer through the miR-181b/HK II axis has been described [56]. However, the authors did not provide any details regarding the potential association between EZH2 and miR-181b.

In addition to post-translational modifications such as ubiquitination, methylation on RNA at the N6 position (m6A) is one of the most prevalent chemical modifications found on mRNAs. This modification has a profound influence on various aspects of RNA metabolism and is regulated by a diverse array of protein molecules. The enzymes METTL3, METTL14, and others are responsible for catalyzing the methylation process on RNA, and recent investigations have uncovered the role of METTL3, a methyltransferase-like 3 enzyme, in contributing to the Warburg effect exhibited in cervical cancer (CC). This effect is achieved by enhancing the stability of HK II through m6A modification, which is facilitated by the m6A RNA methyl recognition protein known as YTHDF1. This finding highlights the significance of m6A modification in cancer biology [57].

As nucleic acid-based therapeutic tools make significant advancements, cyclic RNA has emerged as a potential target molecule for future cancer therapy. However, the precise and efficient knockdown of target genes remains a substantial challenge that needs to be addressed in order to utilize the therapeutic potential of cyclic RNA in cancer treatment to the fullest degree.

### 3.2. HK II Undergoes Downward Adjustment

P-CREB functions as a downstream molecular mediator in AR-mediated glycolysis in hepatocellular carcinoma (HCC), and its silencing effectively abolishes AR-induced glycolysis, cell proliferation, and susceptibility to glycolysis inhibition [58]. Furthermore, Mir-143-3P has shown oncostatic effects across multiple carcinoma types, including lung, prostate, and hepatocellular carcinomas [59,60,61]. Plasma cell tumor variant translocation 1 (PVT1) positively regulates the expression of HK II both in vivo and in vitro by suppressing miR-143 expression [62], and CircRNA-ACAP2 also impedes the invasive, migratory, and anti-apoptotic properties of neuroblastoma cells by targeting the miRNA-143-3-hexokinase 2 axis [63]. miR-216b inhibits tumor development by deactivating the mTOR signaling pathway through targeting HK II [64,65]. Snail also plays a pivotal role in initiating epithelial–mesenchymal transition (EMT) and exhibits high expression in various tumor tissues, such as colon cancer, ovarian cancer, and hepatocellular carcinoma, and as a result, is considered a crucial facilitator of cancer cell invasion and metastasis [66]. 

Suppression of HK II activity using G6P or 2D-G6P mimetics leads to reduced expression of the epithelial-to-mesenchymal transition (EMT) and decreased abundance of Snail proteins. Specifically, increased levels and stability of Snail proteins have been observed as a result of HK II inhibition [67]. HK II interacts with VDAC1 to enable its localization to the mitochondria for glycolysis, and deficiency in sumoylation enhances the binding of HK II to VDAC1 on the outer mitochondrial membrane, thereby promoting glycolysis and facilitating cancer cell proliferation and drug resistance [68,69,70]. Additionally, ubiquitination and subsequent degradation of (HK II) mediated by the ubiquitin-protein ligase E3C are induced by IL13Rα1. IL13Rα1 recruitment and promotion leads to inhibition of glycolysis and promotion of apoptosis in cancer cells [71]. Conversely, knockdown of the KCNQ1OT1 gene in colon cancer (CRC) cells results in the increased ubiquitination of HK II and reduced protein stability [72].

### 3.3. HK Promotes Fructose Utilization

Fructose is a significant source of carbon and energy, conferring an advantageous edge to tumor cells capable of metabolizing it compared to their neighboring counterparts. However, the utilization of fructose for cell proliferation is limited to specific cell types. Recent studies have revealed that the fructose transporter GLUT5 plays a crucial role in facilitating fructose metabolism through glycolysis via hexokinase (HK) rather than ketohexokinase (KHK) [73].

Furthermore, research has highlighted a potential connection between excessive consumption of beverages that contain high-fructose corn syrup (HFCS) and an elevated risk of obesity and colorectal cancer [74]. However, the precise involvement of hexokinase in this relationship necessitates further investigation. Juhong Wang [75] and Michelle K.Y. Siu [76] have made noteworthy contributions in uncovering the non-metabolic functions of HK II, particularly its role in promoting cancer cell stemness. Interestingly, this atypical role of HK II further facilitates cell proliferation, migration, and tumor growth, offering multifaceted insights into its involvement in the progression of tumors. Furthermore, another study conducted by SIRT6 identified a non-metabolic role of HK II in the regulation of genes associated with redox regulation. Notably, HK II was found to have no association with cell death but rather exhibited negative regulation of HIF-1α. Hence, this study emphasized the significance of HK II’s nuclear activity. The above cytokines are summarized in Figure 5.

## 4. The Association between HK II and Tumor Drug Resistance

Tumor resistance mechanisms encompass several aspects, including multi-drug resistance, drug inactivation, alteration of drug targets, drug efflux, DNA repair, inhibition of apoptosis by apoptotic factors, the epithelial–mesenchymal transition, metastasis, the tumor microenvironment, and epigenetic alterations [77,78,79,80,81]. Commonly employed strategies to address these mechanisms involve the combination of drugs, inhibition of drug elimination, modulation of epigenetic factors, and modification of the tumor microenvironment. For instance, the inhibition of drug elimination often targets ABC transporter proteins, which belong to a family that comprises seven subclasses labeled A–G. These proteins play a crucial role in various cellular processes involving the transportation of substances across cell membranes, and several members of this protein family are implicated in drug uptake, excretion, and distribution. When the activity of ABC transporter proteins is hindered, their ability to interact with drugs is inhibited, thereby preventing drug efflux [82].

### 4.1. Mechanisms of Drug Resistance

Hexokinase II expression is up-regulated in tumor cells, granting them metabolic advantages and inhibiting apoptosis. This leads to enhanced cell growth, increasing their resilience against chemotherapy. BAX and BAK are both pivotal apoptotic proteins within cells. They form oligomers and permeabilize the outer mitochondrial membrane (OMM) upon encountering a shift in its retention conformation. Consequently, intermembrane space proteins such as cytochrome c (cyt c) are released, initiating a cascade reaction that involves cysteine asparaginase, which ultimately leads to cell destruction. Anti-apoptotic proteins, including BCL-2, B-cell lymphoma extra-large (BCL-xL), and myeloid leukemia 1 (MCL-1), among others, detect alterations in the protein conformation of BAX or BAK. They transiently interact to revert the translocation of BAX and BAK back into the cytoplasm, effectively averting their activation within the cell, and this mechanism has anti-apoptotic effects [83]. In addition, BCL-2 proteins engage in intermolecular interactions primarily by binding the BH3 structural domain to the hydrophobic groove of another protein [84]. For example, the truncated BH3-only protein BID (tBID) stimulates the activation of BAX and BAK, leading to mitochondrial permeability. Researchers have demonstrated that the direct reverse translocation of hexokinase or HK1/2 selectively reverses tBID entry into the cytosol, thus hindering BAX/BAK activation [85].

Voltage-dependent anion channels (VDACs) feature significantly in mitochondria-mediated apoptosis. The N-terminal structural domain of VDAC1 governs the release of cytochrome c, apoptosis, and the modulation of apoptosis by anti-apoptotic proteins such as hexokinase and BCL-2. These proteins safeguard against apoptosis by interacting with the N-terminal region of VDAC1 [86]. Furthermore, research has revealed that hexokinase plays a crucial role in influencing the functionality of stem cells. Notably, nuclear HK II has been observed to interact with DNA damage response proteins, and the excessive expression of HK II within the cell nucleus has been found to reduce the occurrence of double-stranded DNA breaks while simultaneously enhancing resistance against drugs [87]. This demonstrates the involvement of hexokinase in stem cell biology and highlights potential implications for therapeutic strategies, as illustrated in Figure 6.

#### HK II Mediates Drug Resistance

Twist family bHLH transcription factor 1 (Twist1) is highly up-regulated in various human malignant tumors and plays a crucial role in facilitating the epithelial–mesenchymal transition (EMT) process through multiple signaling pathways [88]. One study has provided evidence of the Twist1-ABCB1 axis involvement in promoting chemoresistance in colorectal cancer (CRC) [89]. Furthermore, researchers have observed that in oxaliplatin-resistant CRC cells, Twist1 is stabilized by HK II via a ubiquitination degradation mechanism, thus exacerbating the progression of EMT and conferring resistance to oxaliplatin [90]. This event further facilitates the progression of the epithelial–mesenchymal transition (EMT) and confers resistance to oxaliplatin treatment in CRC cells.

This impairment, coupled with the activation of the PDK1-HKII-p53 signaling cascade, sustains glycolytic metabolism and gives rise to drug resistance [91]. Additionally, impaired autophagy has emerged as another potential mechanism underlying drug resistance [92,93]. The mTORi pathway, known for its crucial role in autophagy, plays a significant role in many cellular processes. However, when this pathway interacts with HK II, its activity becomes inhibited, causing a reduction in the phosphorylation level of S6K. Consequently, this leads to the development of drug resistance [94]. Additionally, an increased sensitivity to crizotinib has been observed in ALK^+^ NSCLC, which is a type of NSCLC, after inhibiting the AKT/mTOR signaling pathway [95].

Moreover, in trastuzumab-resistant gastric cancer cells, HK II levels were found to be elevated to display a diurnal oscillation pattern. However the viability of trastuzumab-resistant cells can be suppressed through the inhibition of HK II using substances such as 2DG or metformin, ultimately enhancing cell death [96]. As a result, the combination of hexokinase inhibitors with chemotherapeutic agents holds great potential as a robust approach for effectively treating tumors. Collectively, these findings highlight the intricate molecular mechanisms involved in the development of chemoresistance in cancer, and further exploration of these pathways may lead to the identification of novel therapeutic targets for overcoming drug resistance.

### 4.2. Drug Synergism

Unfortunately, the inhibition of specific cellular metabolic pathways is unable to completely impede the proliferation of tumor cells. The presence of metabolic reprogramming in cancer cells allows them to adopt alternative pathways for cellular metabolism, ensuring their survival. Consequently, a more efficacious therapeutic strategy lies in the adoption of a multi-target approach that utilizes drug combinations, instead of solely targeting the glycolytic pathway.

#### 4.2.1. Sorafenib

The multikinase inhibitor Sorafenib exerts multifaceted action by promoting apoptosis, attenuating angiogenesis, and inhibiting tumor cell proliferation. Notably, the silencing of HK II has been found to magnify growth factor deficiencies, thereby making it a potential therapeutic target. Moreover, the combination of HK II inhibitors (3-BrPA, 2-DG) with Sorafenib has shown promising results in enhancing antitumor effects, both in vitro and in vivo. This heightened efficacy stems from a synergistic interplay between the two treatment modalities, as outlined through two distinct mechanisms. First, Sorafenib exposure augments the inhibition of aerobic glycolysis, leading to energy depletion and subsequent apoptosis. Second, Sorafenib induces endoplasmic reticulum stress, further contributing to an increase in apoptosis [97,98].

#### 4.2.2. Rapamycin

Rapamycin, a novel macrolide, is a highly potent and selective inhibitor of mTOR and has received approval from the U.S. Food and Drug Administration (FDA) as an immunosuppressive and anticancer agent. The mTOR pathway serves as a crucial regulatory center for nutrient control that influences the expression of HK II through HIF-1α. This in turn leads to the inhibition of tumor cell growth upon Rapamycin intervention [99]. The co-treatment of 3-BrPA and Rapamycin causes human neuroblastoma (NB) cells to exhibit a substantial increase in apoptosis and also creates [100] a synergistic inhibition of cell proliferation in lung cancer cells [101].

#### 4.2.3. Drugs Related to the Tumor Microenvironment

Tumor hypoxia constitutes a prominent aspect of the tumor microenvironment that primarily arises from an imbalance between a diminished oxygen supply due to abnormal vascularization and a heightened oxygen consumption by tumor cells. Hypoxia engenders intensified drug resistance and triggers the induction of autophagy in tumor cells under hypoxic stress, and this poses a formidable challenge in treating many types of cancer. Some researchers have proposed that patients who receive autophagy inhibitors as monotherapy and do not exhibit a sufficient response may benefit from autophagy inhibitors with glycolysis inhibitors [102]. For instance, the co-administration of 3-BrPA and Bortezomib notably amplified apoptosis in hypoxic myeloma cells in one study [103]. Moreover, it is plausible that HK II plays a role in regulating autophagy activation in hypoxic myeloma cells. Due to an inadequate oxygen supply, tumor cells engage in energy metabolism predominantly through anaerobic fermentation (the Warburg effect). Consequently, lactic acid accumulates, and in an attempt to prevent intracellular acidosis, ion exchange proteins on the tumor cell membrane actively transport H^+^ ions out of the cell. These cellular processes contribute to a decrease in pH within the tumor microenvironment, making it acidic. The acidification caused by lactic acid not only induces apoptosis in neighboring healthy cells but also facilitates the development of low pH-resistant cancer cells. Moreover, lactic acid influences the adhesion of tumor cells to the extracellular matrix by regulating interactions with integrins and stimulating the secretion of matrix-degrading enzymes, thereby promoting cell metastasis. The acidic pH environment also fosters an immunosuppressive microenvironment, thereby promoting immune evasion by tumor cells [104,105,106]. Additionally, hepatocellular carcinoma (HCC) is characterized by the up-regulated expression of carbonic anhydrase IX (CA-IX), an enzyme implicated in pH reduction, and in recent years, carbonic anhydrase inhibitors have emerged as potential agents for hypoxic tumor therapy and imaging [107]. Combining CA-IX inhibitors, such as acetazolamide, or silencing CA-IX with 3-BrPA, has also shown promise in augmenting tumor apoptosis [108,109].

#### 4.2.4. Oncolytic Virus Therapy

After the groundbreaking discovery by Martuza et al. in 1991 [110] that transgenic HSV holds promise in the treatment of malignant glioma, the field of oncolytic viral therapy that utilizes HSV has garnered significant attention. This therapy involves the creation of modified lysogenic viruses with weak pathogenicity through genetic engineering that can selectively infect tumor cells. Once inside, these viruses vigorously replicate, leading to the eventual destruction of the tumor cells. In addition, this therapy elicits an immune response, attracting immune cells that aid in the eradication of residual cancer cells. Notably, the combination of D-mannoheptulose and Newcastle Disease Virus (NDV) has been found to effectively impede glycolytic products in treated breast cancer cells while sparing normal cells. Moreover, the presence of the hexokinase inhibitor D-mannoheptulose augments the sensitivity of breast cancer cells to the tumorolytic effects of Newcastle Disease Virus [111]. However, a separate investigation revealed that the suppression of HK II or the concomitant use of 2-DG resulted in a reduction in the infectious and tumorigenic impact of M1 viruses, indicating that the glycolytic pathway plays a crucial role in facilitating the infectious and tumorigenic effects of these viruses. Conversely, lonidamine exhibited the opposite effect by augmenting the infectious and tumorigenic potential of M1 viruses through an alternative mechanism that operates independently of HK II [112].

#### 4.2.5. HDACis

Hexokinase types I and II are localized in the outer mitochondrial membrane, enabling them to access ATP generated by the mitochondria by binding with VDAC. This interaction promotes glucose phosphorylation, thereby facilitating tumor cell proliferation and metastasis. Although histone deacetylase inhibitors (HDACis) have exhibited success in treating T-cell lymphomas, their effectiveness in solid tumors has been limited. However, one promising approach involves combining azole inhibitors clotrimazole and bifoncarbazole, which disrupt mitochondrial hexokinase activity, with romidepsin, an HDACi. This combination enhances the efficacy of HDI in the treatment of solid tumors [113]. Moreover, proteomic studies have demonstrated the effectiveness of a novel drug combination comprising HDACis and 2-DG in various solid tumor cell lines [114].

Otherwise, in a study conducted by Sameer Agnihotri et al., the screening of azole antifungal drugs revealed their potential to impede tumor metabolism. Notably, ketoconazole and posaconazole demonstrated the most robust inhibitory effects on glioblastoma (GBM), as observed both ex vivo and in vivo [115]. These findings offer valuable mechanistic insights in support of the integration of azoles in GBM treatment approaches.

#### 4.2.6. Alkylating Agents

Alkylating agents exert their antitumor effects by forming covalent bonds with macromolecules such as DNA, RNA, and proteins. The alkylation reaction leads to the formation of crossovers or depurinations within these macromolecules, ultimately disrupting their normal functions. However, the intricate biological processes associated with tumor development allow tumor cells to evade the cytotoxic effects triggered by alkylators. One notable mechanism involves the utilization of methylguanine (MGMT) to repair DNA damage, wherein the detoxification activity of MGMT helps tumor cells evade cell death induced by alkylating agents. Inhibition of MGMT effectively enhances the sensitivity of tumor cells to antitumor drugs [116,117]. In the study conducted by Sun et al., the co-administration of HK II inhibitors such as 2-DG or 3-BrPA augmented the sensitivity of tumor cells to BCNU when compared to the usage of chloroethylnitrosoureas (CENUs) alone [118,119,120]. Although the correlation between MGMT and HK II remains unknown, these findings have significant implications for the enhancement of alkylating agents in antitumor therapies.

#### 4.2.7. Other Antitumor Drugs

Metformin, a widely prescribed first-line treatment for type II diabetes, has demonstrated its efficacy beyond diabetes management and has emerged as a promising option in cancer therapy. In addition to its established role with type II diabetes, metformin has shown significant potential in sensitizing cancer cells, thus enhancing the effectiveness of cancer treatment. The triple combination of HK II’s antisense oligonucleotide, metformin, and pipecillin has exhibited synthetic killing in cultured multiple myeloma cells, while also impeding the progression of HK1-HK II^+^ multiple myeloma tumors in experimental models [121]. Studies have also found that metformin acts as an inhibitor of HK II, further supporting its significance in cancer treatment [122].

In the realm of therapeutics, arsenic and its derivatives have been widely employed for the treatment of various diseases, with acute promyelocytic leukemia (APL) being one of the most successful applications. Notably, a study conducted by Zhang et al. revealed that arsenic trioxide (As_2_O_3_) significantly inhibits the activity of HK II. Since HK II is considered the rate-limiting factor in the glycolytic pathway, a crucial energy metabolism pathway in cancer cells, by disrupting cellular metabolism, arsenic trioxide ultimately induces apoptosis, leading to cell death [123]. Furthermore, researchers have observed that arsenic trioxide also counteracts tamoxifen resistance, another significant finding with potential implications for breast cancer treatment [124]. Arsenic trioxide has thus emerged as a valuable asset in the realm of synergistic antitumor therapeutic strategies, and the combination of metformin and arsenic trioxide has proven to be particularly effective as metformin sensitizes hepatocellular carcinoma to arsenic trioxide-induced apoptosis while simultaneously inhibiting proliferation by down-regulating Bcl2 expression [125]. 

One proteomic investigation also uncovered that metformin enhances the sensitivity of intrahepatic cholangiocarcinoma to arsenic trioxide, thereby impeding proliferation through modulation of the AMPK/p38 MAPK-ERK3/mTORC1 pathway [126]. Moreover, Jiao Wang’s research indicates that metformin exhibits dual functionality by inhibiting HK II and inducing the degradation of the PER1 protein in both human cells and live mice. This disruption of circadian rhythms has been found to counteract tricholoma resistance [96]. Finally, the neo-adjuvant treatment of HER2-positive breast cancer has shown promise in phase II studies with the combination of metformin, liposomal adriamycin, docetaxel, and trastuzumab [127], as illustrated in Table 1.

## 5. HK II Inhibitors

2-Deoxyglucose (2-DG), 3-Bromopyruvate (3-BrPA), lonidamine (LND), and metformin are inhibitors of hexokinase II (HK II) commonly utilized in experimental studies. Nevertheless, the clinical trials involving these compounds have sparked controversy, as illustrated in Figure 7. For instance, 2-DG was withdrawn from clinical use as an antitumor drug due to its limited tolerability and side effects, which inadvertently activated pro-survival pathways in cancer cells. However, it is currently being investigated in phase II clinical trials for its potential in treating epilepsy [128,129,130,131]. The limitations of 3-BrPA are as follows: its rapid inactivation or resistance in glutathione-rich tumors, nonselective alkylation properties, potential for off-target interactions with unknown proteins, and challenges associated with penetrating the blood–brain barrier for treating gliomas [132,133,134,135]. LND, a selectively active compound against a wide range of tumors that is approved for use in certain countries, exhibits limited efficacy as a standalone chemotherapeutic agent for inhibiting cancer cell growth, both in vivo and in vitro [136]. However, when combined with other chemotherapeutic agents, LND demonstrates nonoverlapping toxicities [137]. Consequently, LND is frequently employed as a chemosensitizer. Similarly, metformin possesses relatively modest antitumor activity and cannot be relied upon as a standalone antitumor agent [138].

In an effort to identify potent inhibitors of HK II, Liu employed virtual screening of structure-based zinc databases and identified two classes of small molecule inhibitors: benzylhydrazines and arylhydrazines. Among these, Benitrobenrazide (BNBZ) exhibited the highest activity and a reasonable selectivity for HK II. Many experiments have also demonstrated its antitumor inhibitory ability in vivo and ex vivo, but it is currently only being used in experimental studies and has not entered into clinical trials [139,140,141].

VDA-1102, a novel and potent selective mutational modulator of HK II, exhibits the ability to dissociate from cancer cell mitochondria as well as certain immune cells. Through this dissociation, VDA-1102 effectively triggers apoptosis in cancer cells by disrupting the mitochondrial association with HK II. Furthermore, it exerts inhibitory effects on glycolysis, mitigates immunosuppression within the tumor microenvironment, and stimulates antitumor immune responses. Actinic keratosis (AK), a globally prevalent skin disease, represents an early stage of cutaneous squamous cell carcinoma (cSCC), and VDA-1102 is being developed as a topical ointment for the treatment of AK and other non-melanoma skin cancers. It selectively facilitates the dissociation of HK II from VDAC1 in a dose-dependent manner. Notably, a phase II clinical trial [142,143] has been successfully conducted to evaluate its efficacy. Sheng H. published a comprehensive review on inhibitors that target key enzymes involved in the glycolytic pathway [139] that played a pivotal role in initiating research focused on developing small molecule inhibitors that target hexokinase.

One natural inhibitor of HK II is glucose-6-phosphate (G6P), which is produced as a byproduct of HK II-catalyzed glucose metabolism, and previous studies have demonstrated that G6P facilitates the translocation of HK II from the mitochondria to the cytoplasm [144]. For example, Wenying Shan et al. employed docking studies to investigate the interaction between G6P and HK II. Their research showed that G6P exhibits specific polar interactions with three key residues, namely ASP447, SER449, and LYS451, located within the N-terminal catalytic domain of HK II, as well as with the linker helix α13. The identification of key fragments that can bind to the G6P-binding site has revealed several promising candidates, including sulfonamides, polyhydroxypyrans, lactam heterocycles, polycyclic and multipolar moieties, and lipophilic fragments such as phenyl acyl or cycloalkyl groups. These fragments all demonstrate a reduction in molar polarity, thereby enhancing permeability and bioavailability [145,146]. Utilizing these active fragments and their bio-identicals, G6P mimetics can be designed to target HK II specifically. 

Furthermore, the inhibitor trehalose-6-phosphate (T6P), originally identified as a yeast HK II inhibitor, has been found to effectively inhibit human HK II while not affecting HK I. However, due to the low cellular uptake of phosphorylated sugar and potential inaccuracies in toxicity testing, there is a need to improve uptake through alternative formulation techniques [147]. These findings shed light on the structural mechanisms that govern the inhibitory action of G6P and provide valuable insights into the functional interplay between G6P and HK II.

### 5.1. Natural Compounds

Plants represent an invaluable source of natural medicines, encompassing not only their prominent role in traditional Chinese medicine but also the significant value derived from the isolation of effective plant-derived compounds. In China, Wang Pu conducted an extensive analysis of existing research concerning the utilization of natural products to overcome tumor resistance [148]. Experimental evidence has demonstrated the antitumor effects of Epicatechin gallate (EGCG) and quercetin [149]. Furthermore, Asifa Khan employed molecular docking techniques to identify Epicatechin gallate (EGCG) and quercetin as compounds with remarkable affinity, efficiency, and specificity for the HK II-binding pocket, which are all favorable drug-like properties [150]. Hence, these compounds may be able to serve as potential inhibitors of HK II. Another significant constituent, diosgenin, found in Dioscin, functions by enhancing the binding of the E3 ligase FBW7 to c-myc. This in turn promotes c-myc ubiquitination, leading to its degradation and subsequently inhibiting HK II. Additionally, diosgenin can mitigate the interaction between HK II and VDAC-1, ultimately inducing apoptosis [151]. 

Matrine, an alkaloid extracted from Panax ginseng, also exhibits the ability to suppress HK II expression at a concentration of approximately 2.0 μM. This effect is achieved by reducing the binding of c-myc to the introns of the HK II gene. Moreover, when used in combination with the HK II inhibitor LND, matrine demonstrates a synergistic effect in the treatment of myeloid leukemia [152]. First, Geeta Swargiary et al. conducted molecular docking analysis, using G6P as a control, to dock natural compounds at the G6P-binding site of HK II, screening out 11 natural compounds. Based on binding energy, the top three compounds, Bayogenin (−7.2 Kcal/mol), asiatic acid (−7 Kcal/mol), and andrographolide (−6.9 Kcal/mol), were selected for molecular dynamics simulations (MDS) and confirmed that these compounds may be better and potential ligands for HK II, serving as inhibitors of HK II in cancer cells [153]. It was found that andrographolide could potentially have a similar effect to G6P on HK II, indicating the potential of andrographolide to inhibit HK II by occupying the G6P-binding site. Specifically, asiatic acid and bayogenin obtained from CA, as well as andrographolide derived from AP, have been proven to have antitumor effects [154,155,156,157], but their relationship with HK II needs further validation through in vitro and in vivo experiments.

Pachymic acid (PA) is a triterpene found in Poria, and molecular docking revealed that PA has a much higher affinity for HK II than G6P (molecular docking scores of 8.18 and 5.31, respectively). PA was found to significantly inhibit HK II activity in cell lysates (IC50 5.01 μm), induce mitochondrial dysfunction, ATP depletion, and ROS generation. It can also act as a competitive activator of fructose-1,6-bisphosphate as an allosteric activator of pyruvate kinase M2 (PKM2) [158].

Earlier, we mentioned that HIF-1α is a positive regulator of HK II. Research has found that α-Hederin can inhibit c-Myc and HIF-1α by activating SIRT6 expression, thereby suppressing glycolysis-related enzymes including HK II, PKM2, and LDHA [159].

Glycyrrhizic acid (GA) is a significant constituent of licorice root that has been found to impede the proliferation of both LC cells and glycolytic metabolism. Sun’s study, showed that the proliferation of LC cells can be significantly stimulated when the expression of HK II increases, while GA (100 µg/mL) can inhibit this cell proliferation. It was also mentioned in the previous discussion that AKT can promote the expression of HK II, and in this experiment, an increase in the phosphorylation level of AKT was found in LC cells. However, after exposure to GA, the phosphorylation level of AKT decreased. Therefore, it can be concluded that GA can inhibit HK II through the PI3K/AKT pathway, thus inhibiting the proliferation and glycolysis metabolism of LC cells [160]. In addition, 13 steroidal compounds were isolated from the fruiting bodies of Ganoderma lucidum, and a new steroid compound (22 E, 24 R)-6β-methoxyergosta-7,9(11),22-triene-3β,5α-diol(2) with the highest binding affinity for HK II was discovered. In vitro studies further confirmed compound **2** as a potential HK II inhibitor [161].

In addition to plants, actinomycetes derived from insects have also emerged as a promising reservoir of novel natural products. A particular genus of streptomycetes has been discovered within the larvae of mud wasps that has led to the isolation of three novel compounds known as streptantibins A-C from ethyl acetate extracts. Notably, these compounds have exhibited considerable inhibitory activity against HK II in vitro, as evidenced by their IC_50_ values of 9.8, 34.6, and 31.2 Μm, and showed low toxicity to human normal liver cells, making them potential lead compounds [162].

Thymoquinone (TPL) is a natural product isolated from traditional herbal medicine, and research has found that TPL eliminates head and neck cancer cells through inducing pyroptosis mediated by gasdermin E (GSDME). In this experiment, the authors found that TPL treatment up-regulated the protein levels of BAX and BAD in HK1 and FaDu cells, while not affecting the protein level of BAK1, as determined by Western blot experiments. As discussed earlier, we know that HK II can inhibit apoptosis induced by BAX/BAK and cytochrome c, so the authors found through RNA-seq that hexokinase II (HK-II) is down-regulated by TPL, indicating TPL’s inhibitory effect on HK II [163]. The Hedgehog (Hh) signaling pathway controls cell fate, proliferation, and differentiation, and when this pathway is abnormally activated, it can lead to the development of tumors. In the study by Ling Li et al., it was found that wogonin can inhibit the Hh pathway, as well as its inhibitory effect on several key enzymes controlling glycolysis (HK II, PFK2, GLUT1, and MCT-4), providing further evidence of the connection between glycolysis and Hh signaling regulation [164,165,166]. Methyl jasmonate (MJ) is a hexokinase-II inhibitor that can inhibit tumor cell proliferation by blocking glycolysis through disrupting the interaction between VDAC and HK-II on the mitochondrial membrane. Bilgesu Onur Sucu modified the ester portion of MJ’s structure using an oxadiazole moiety to obtain a more effective HK II inhibitor than the original (IC_50_: compound 3 = 0.27 µM, MJ = 7.47 µM) [167]. Another example is Sulforaphane, which is derived from the hydrolysis of thioglycosides by black mustard enzyme and is a sulfur-containing compound found in cruciferous plants such as broccoli, kale, and northern round carrots. Huang et al. found that in mice and bladder cancer cells induced by N-butyl-N-(4-hydroxybutyl) nitrosamine, sulforaphane can down-regulate the protein and gene expression of HK II, PKM2, and PDH, as well as inhibit the expression of AKT1. That is, sulforaphane inhibits glucose metabolism mediated by the AKT1/HK II axis, and treats bladder cancer [168]. In addition, studies have found that the metabolites of quercetin, curcumin, and ginsenosides inhibit glycolysis by inhibiting the AKT/mTOR signaling pathway [169,170,171,172]. These findings shed light on the potential of actinomycetes from insects as a valuable source for the discovery of bioactive compounds with diverse therapeutic applications.

### 5.2. Targeted Drugs

Mitochondria regulate a substantial negative transmembrane potential along their inner membrane via the electron transport chain (ETC). This distinct physical attribute facilitates the targeting of mitochondria by utilizing positively charged portions. Peptides possess numerous desirable attributes, such as cost effectiveness, straightforward production, biocompatibility, biodegradability, controllable size, structural adaptability, and versatile functionality, and as a result, have emerged as a promising avenue for therapeutic interventions and as a means to facilitate drug delivery. The cellular uptake, mitochondrial localization, and cytotoxicity of a peptide in HeLa cells have been found to be significantly enhanced through covalent coupling of a short penetration-accelerating sequence (PAS). Abiy D. Woldetsadik developed a cell-penetrating peptide (pHK-PAS) by linking the 15 amino acids of the N-terminal end of the mitochondrial-membrane-bound HK II (pHK) (HKII(pep)) with the penetration-accelerating sequence (PAS). This peptide disrupted the mitochondria-HK II linkage in cancer cells, resulting in mitochondrial dysfunction and eventual apoptosis [173].

Karolina Kozal developed a cancer-selective cell-penetrating peptide (CPP) that, when used alone or in combination with miR-126, specifically displaces HKII in the mitochondria of cancer cells. Importantly, this CPP does not affect the response of the voltage-dependent anion channel (VDAC) to the outer mitochondrial membrane (OMM) localization. Consequently, it triggers the release of cytochrome c to the cytoplasm, leading to apoptosis [174].

Another study on peptides was carried out by Zhang Meng et al. who employed a targeting strategy using the follicle-stimulating hormone (FSH) peptide. They loaded HK II shRNA onto a PEG-PEI copolymer that was conjugated with FSH peptide along with reverse-transcribed FSH peptide, and this approach selectively inhibited the expression of HK II in ovarian cancer [175]. However, the application of this drug class still faces certain obstacles, including a short in vivo half-life attributed to protease degradation and rapid renal clearance, as well as low targeting efficiency.

Multi-walled carbon nanotubes (MWCNTs) also hold significant potential for a broad range of applications in various fields. However, they also exhibit notable biotoxicity. Specifically, peptide-based multi-walled carbon nanotubes (MWCNTs) represent a promising avenue for drug delivery by exploiting the lipophilic cation Rho-110 for mitochondrial functionalization, thereby directing their action towards the mitochondria [176,177,178]. HkII(pep), when coupled with peptide-based multi-walled carbon nanotubes (MWCNTs), exhibits enhanced cell permeability of HkII(pep), leading to apoptosis promotion [179]. Utilizing peptide-based approaches for mitochondrial targeting provides valuable insights into mitochondrial drug treatment and potential drug targets.

In another investigation, a glucose-targeted carrier was developed that was found to selectively aggregate in the hypoxic region of tumor spheroids and facilitate the controlled release of anticancer drugs specifically in the tumor microenvironment. This approach reduced the required drug dosage, minimized side effects, and incorporated the use of biologically inert metal complexes, specifically Co (III), as a synergistic drug chaperone for tumor activation, thus improving the therapeutic index compared to single glucose vector (glucose is conjugated with tris(methylpyridyl)amine) [180]. These findings offer novel perspectives in the design of cancer chemotherapy studies.

Enping Chen presented a notable study on the development of a mitochondria-targeted pH-sensitive poly (vinyl alcohol) (PVA) nanogel. This innovative nanogel formulation effectively combined the hexokinase inhibitor LND and the chemotherapeutic drug paclitaxel (PTX) to address drug resistance in tumor treatment [181]. The objective of restoring the activity of paclitaxel and achieving a synergistic effect for the treatment of drug-resistant tumors was achieved through the accumulation of the nanogel in the mitochondria. The released LND, on the one hand, the inhibition of mitochondrial membrane potential directly induced cell apoptosis and promoted oxidative stress generated by ROS. On the other hand, it sensitized drug-resistant tumors by blocking the energy source and promoting the reversal of drug resistance and the sensitization of PTX activity.

### 5.3. Residue Targets

In contrast to the C domain, the N domain (NTD) of HK II plays a crucial regulatory role in both enzyme stability and the inhibition of apoptosis [12]. A specific residue in the NTD, G231D, undergoes mutation that results in the destabilization of HK II and the near total loss of its catalytic activity [182]. The activity of the NTD is impacted by the size of the linker helix-α13, which serves as a connection between the N-terminal and C-terminal structural domains of HK II. A study conducted by Juliana C. Ferreira et al. uncovered that three inactive site residues located at the initial segment of the linker helix-α13 (D447, S449, and K451) regulate the NTD activity of HK II. Moreover, they proposed that these residues serve as a regulatory site for HK II, which could be targeted to specifically inhibit the enzyme. This inhibition, in turn, could effectively reduce the rate of cancer glycolysis, offering a promising avenue for the development of novel anticancer drugs [183]. Additionally, in a study conducted by Ruiqi Li et al., mutations such as V431Cfs*26, L795Rfs*10, and A901Rfs*74 were found in cases of gastric adenocarcinoma (STAD), lung squamous carcinoma (LUSC), and endometrial carcinoma (UCEC). However, the study did not establish a statistically significant association between the mutations in the HK II gene and the survival rates of different tumor types [10].

### 5.4. Lead Compounds

According to the estimates of cancer incidence and mortality compiled by the International Agency for Research on Cancer (IARC) in 2020, the latest information on the global burden of cancer has been provided. It is predicted that there were approximately 19.3 million new cancer cases worldwide in 2020 and approximately 10 million cancer deaths. And compared to 2020, the number will increase by 47% by 2040. [184]. Inhibitors of HK II such as 2-DG, 3-BP, and LND have shown good activity in experiments, but due to their side effects and other factors, there are currently no HK II inhibitors on the market. Therefore, it is an important issue to efficiently search for new HK II inhibitors. In recent years, with the development of computer technology and bioinformatics, it has become possible to search for new HK II inhibitors through computational chemistry methods. By using molecular simulation, potential HK II inhibitors can be predicted and further experimentally validated, significantly speeding up the experimental process.

The utilization of high-throughput virtual screening represents an important approach in the discovery of novel drugs, especially as it relates to lead and other metallic compounds. Rui Shi successfully identified compound 27 as the most potent inhibitor of HK II, showcasing its remarkable efficacy against glioma cells amidst a vast pool of 240,000 compounds [185]. Compared to 3-bp, its activity has improved five-fold (U87 glioma cells, 11.3 μM for Compd 27; 62.4 μM for 3-BP). Furthermore, this inhibitor exhibited favorable blood–brain barrier penetration and demonstrated minimal adverse effects on normal tissue. Another study found that Benserazide (Benz) is a dopamine decarboxylase inhibitor used in the treatment of Parkinson’s disease and Parkinson’s syndrome. Additionally, research has found that Benserazide is a selective HK II inhibitor, which can specifically bind to HK II and significantly inhibit HK II enzyme activity in vitro [186]. Therefore, the author further screened a library of an extensive library housing over 6 million zinc and lead-like compounds, leading to the discovery of a novel scaffold for HK II inhibition, namely (E)-N-(2,3,4-trihydroxybenzyl) arylhydrazide. This compound, known as Benitrobenrazide, directly targeted HK II, resulting in the inhibited proliferation of cancer cells. Whether in extracellular enzyme inhibition experiments or cellular enzyme inhibition, Benitrobenrazide exhibits higher activity than LND, 2-DG, and 3-BP (IC50_HK II_ = 0.53 μM, IC50_SW1990_ = 24 μM; IC50_SW480_ = 7.13 μM). It also displayed high selectivity and potent inhibitory activity against HK II [139], as illustrated in Figure 8.

Schiff base compounds, which encompass heterocyclic ring systems such as quinazolinones and indole rings, as well as azomethine bonds, have garnered considerable interest for their promising applications in the fields of medicine and medicinal chemistry. Notably, Ahmed A. Noser has undertaken the design and synthesis of novel compounds: compound **11** (a Schiff base formed from quinazoline amino acids) and compound **20** (a Schiff base derived from indole amino acids). These compounds have demonstrated the ability to induce apoptosis and cellular death by triggering the generation of reactive oxygen species (ROS), thereby inhibiting mitochondrial complex I-related hexokinase [187]. As illustrated in Table 2.

## 6. Conclusions

Metabolic reprogramming plays a vital role in facilitating the growth, proliferation, and survival of cancer cells. Intrinsic mechanisms within these cells trigger the activation of signaling components that have a direct impact on enhancing the activity of metabolic enzymes or up-regulating transcription factors. Consequently, this leads to an increase in the expression of metabolic regulators [188,189].

The organism is a complex regulatory process, and proteins or transcription factors such as ErbB2 and BACH1, which are involved in tumor development, also have positive/negative regulation on HK II. The Wnt/β-catenin signaling pathway is a very important pathway in biology, playing a crucial role in embryonic development and tissue regeneration. However, when it is abnormally activated, it can lead to the occurrence of cancer, including affecting the expression of downstream proteins such as cyclin D1 and c-myc and MMPs (matrix metalloproteinases) [190,191]. HK II acts as an upstream protein of Wnt/β-catenin and regulates it. In addition, HectH9 regulates the stability of HK II through ubiquitination modification or M6A, thereby affecting the expression of HK II. Non-coding RNAs have diversity, precision, reversibility, and other advantages in regulating protein expression, making them an indispensable part of the cellular regulatory network. Experimental evidence has shown that they directly regulate cellular functions, such as the expression of HK II, by binding to proteins and mRNA, thus controlling the progression of cancer [192,193].

Drug resistance is a critical determinant of drug efficacy, prominently witnessed in the context of drug activity. Pertinently, the inhibition of apoptotic factor production by HK II serves as an instrumental factor in engendering cellular resistance. In this vein, the concomitant administration of HK II inhibitors alongside antitumor medications manifests as an efficacious strategy to curtail cellular resistance, thereby endowing a dual antitumor effect.

Natural compounds play a crucial role in the discovery of novel drugs due to their distinctive chemical structures and pharmacological activities. High-throughput screening expedites the identification of potential structures, resulting in significant time and cost savings. Precise drug development is enhanced by targeting specific residues [183], while drug repositioning and formulation modification are effective strategies to discover new drugs and minimize expenses and time consumption.

One such regulator of particular interest is HK II, a member of the hexokinase family, because it is highly responsive to the demands for biomolecules and energy required by cancer cells to support their growth and proliferation. Furthermore, HK II plays a significant part in modulating anti-apoptotic mechanisms within cancer cells. Given HK II’s pivotal role in regulating tumor development, the exploration of novel antitumor agents targeting HK II holds considerable promise.

## Figures and Tables

**Figure 1 molecules-29-00075-f001:**
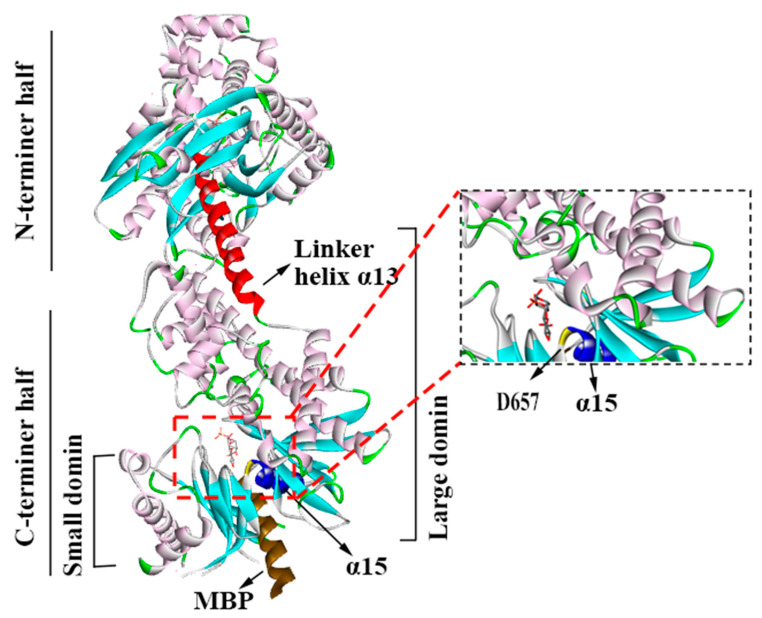
The structure of HK II (2NZT, PDB). HKⅡconsists of N- and C-terminal halves, and they are joined by *α*13. Each half also comprises large and small domains.

**Figure 2 molecules-29-00075-f002:**
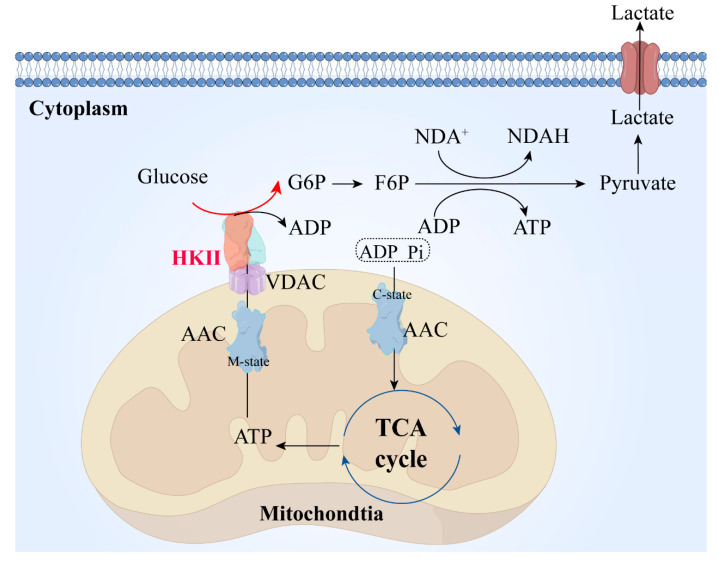
HK II during glycolysis. ADP and PI in the cytoplasm are transported into the mitochondrial matrix through the mitochondrial inner membrane via the AAC protein. They participate in the TCA cycle. The ATP produced is transported from the mitochondrial matrix to the intermembrane space through the AAC protein and participates in the process of glucose conversion to G6P mediated by HK II. The lactate produced is transported out of the cell through transport proteins on the cell membrane. AAC, mitochondrial ADP/ATP carrier; m-state, matrix-open state, the AAC protein transports ADP into the mitochondrial matrix, with the structure of the AAC protein referred to as the m-state during this process; c-state, cytoplasmic-open state, the AAC protein transports ADP into the mitochondrial matrix, with the structure of the AAC protein referred to as the c-state during this process; TCA, tricarboxylic acid cycle; VDAC, voltage-dependent anion-selective channel; HK II, hexokinase II; G6P: glucose 6-phosphate; F6P, fructose 6-phosphate; NDA+, nicotinamide adenine dinucleotide; NDAH, nicotinamide adenine dinucleotide (prototype).

**Figure 3 molecules-29-00075-f003:**
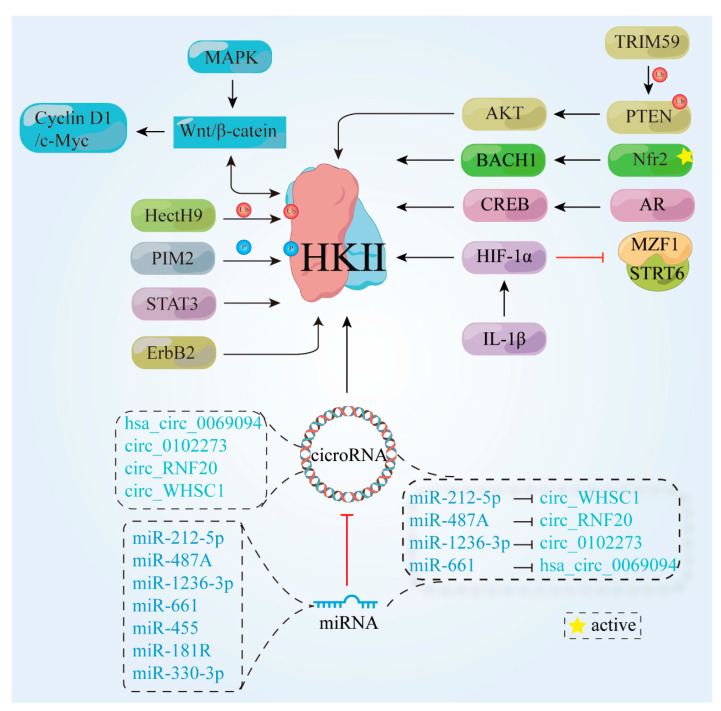
Proteins and signaling pathways that cause HK II up-regulation. ErbB2, tyrosine kinase receptor 2; HectH9, an enzyme belonging to the HECT (Homologous to the E6-AP Carboxyl Terminus) protein family; TRIM59, tripartite motif-containing protein 59; PTEN, phosphatase and tensin homolog; AKT, protein Kinase B; BACH1, BTB and CNC homology 1; Nrf2, nuclear factor erythroid 2-related factor 2; PIM2, pre-B lymphoma virus insert site 2; MAPK, mitogen-activated protein kinase; STAT3, transcription 3; Wnt/β-catenin, canonical Wnt/β-catenin pathway; C-myc, cellular-myelocytomatosis viral oncogene; Cyclin D1, G1/S-specific cell cycle protein-D1; CREB, cAMP response-element binding protein; AR, androgen receptor; HIF-1α, hypoxia inducible factor 1 subunit alpha Gen; MZF1, myeloid zinc finger protein; SIRT6, a member of the histone deacetylase family; IL-1β, Interleukin-1β; 
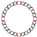
, cicroRNA; 
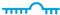
, miRNA; 
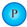
, phosphorylation; 
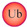
, ubiquitination; 
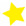
, active.

**Figure 4 molecules-29-00075-f004:**
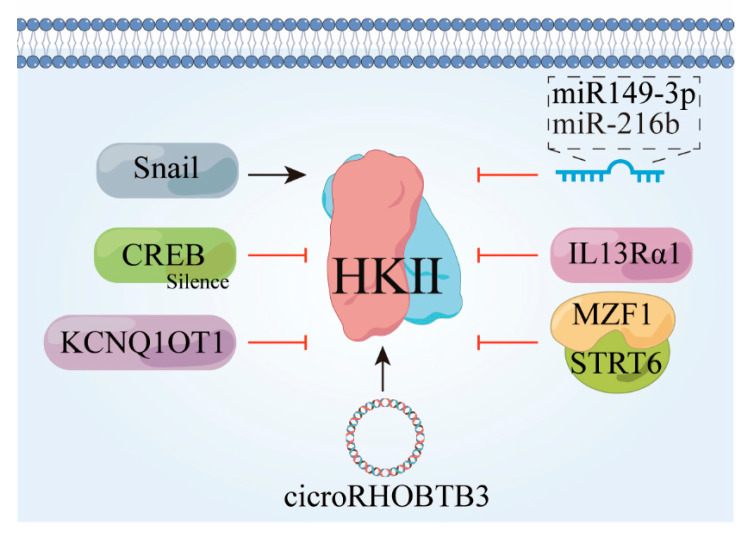
Factors that contribute to the downward revision of HK II. Snail, snail family transcriptional repressor 1 gene; CREB, cyclic-AMP response-binding protein; KCNQ1OT1, a long non-coding RNA; IL13Rα1, interleukin-13 receptor subunit alpha-1.

**Figure 5 molecules-29-00075-f005:**
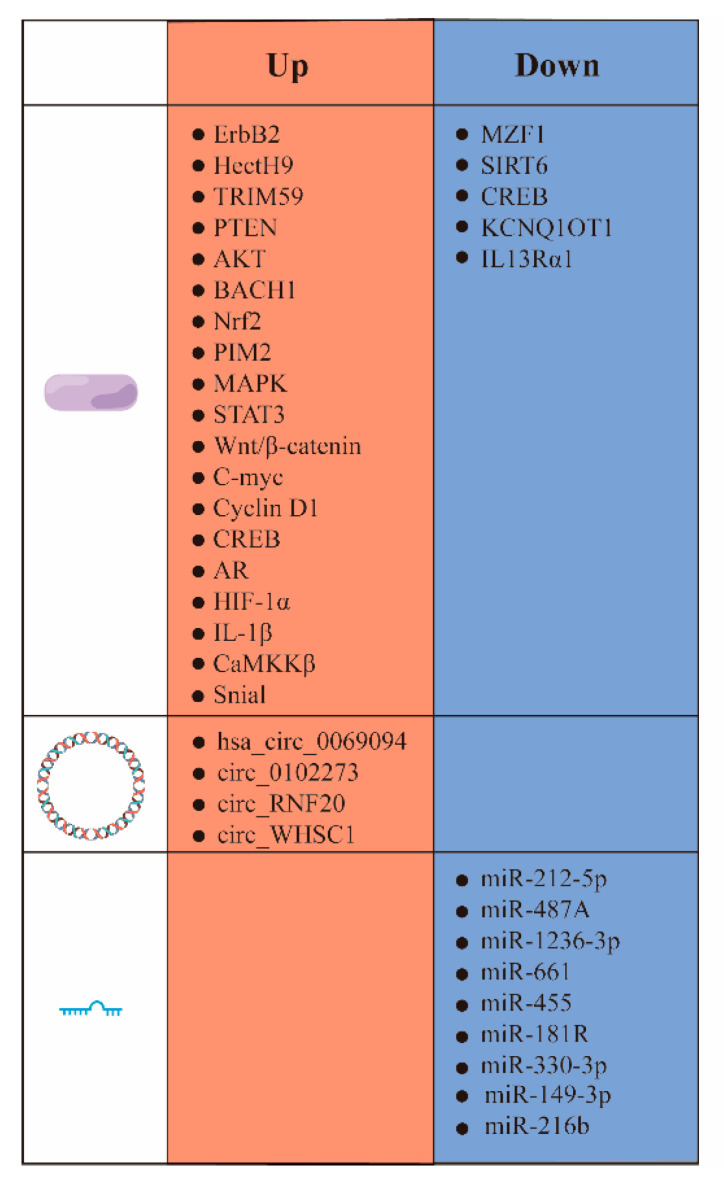
Summary of various biological factors that impact HK II expression. 
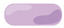
, protein; 
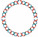
, cicro RNA; 
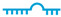
, miRNA.

**Figure 6 molecules-29-00075-f006:**
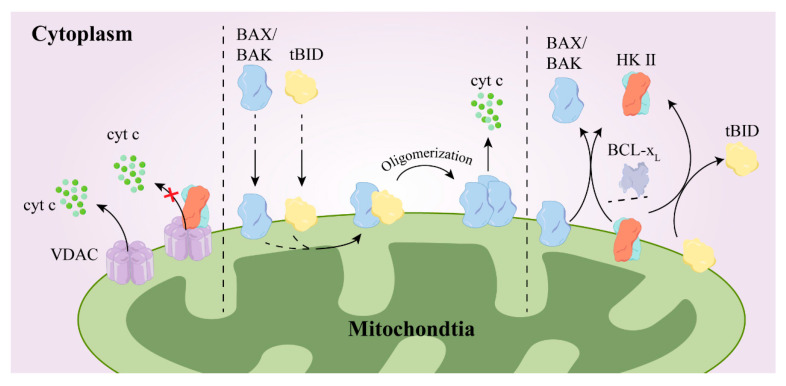
The mechanism by which HK II promotes drug resistance. After binding with VDAC, BCL-2 and/or HKII inhibit the release of cytochrome c from VDAC (left); when BAX and tBID are in the outer mitochondrial membrane, tBID binds to BAX/BAK, promoting their oligomerization and the release of cyt c (medium); HK II and BCL-x_L_ reverse the translocation of tBID and BAX/BAK from the outer mitochondrial membrane to the cytoplasm (right). BCL-2, B-cell lymphoma-2; BAX/BAK, BCL2-Associated X/K; Tbid, Bcl-2/BH3-only protein; **×**, means inhibit.

**Figure 7 molecules-29-00075-f007:**
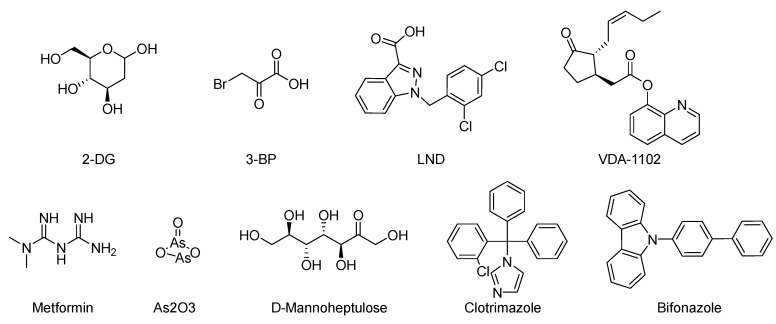
Commonly used HK II inhibition in recent studies.

**Figure 8 molecules-29-00075-f008:**
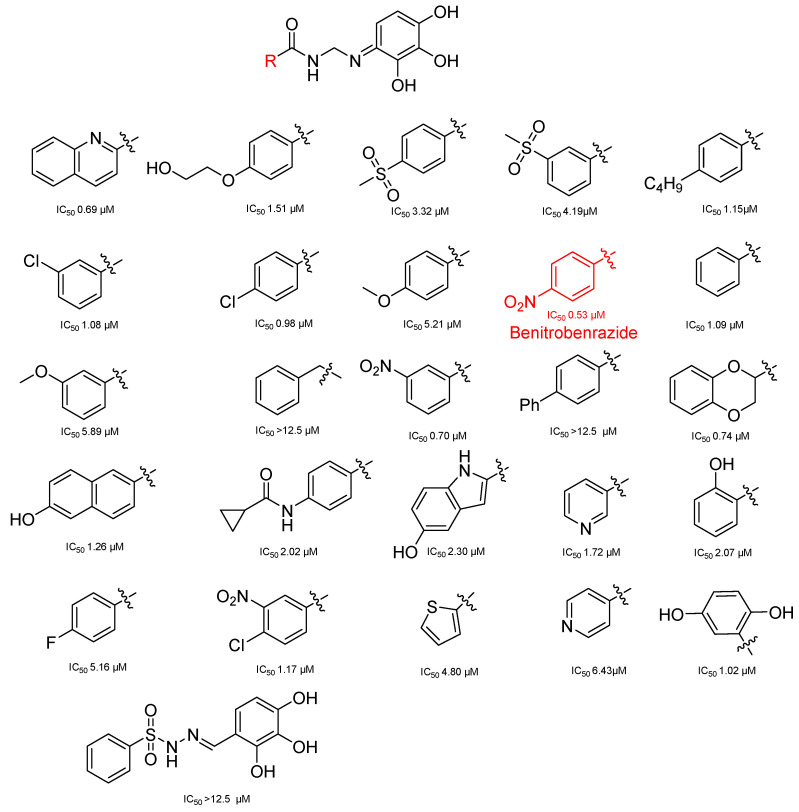
N-(2,3,4-trihydroxybenzyl)arylhydrazide HK II inhibitors and their conformational relationships. The highlighted structure is the most active.

**Table 1 molecules-29-00075-t001:** Synergistic effects of antitumor agents.

Number	Antitumor Drug	Mechanism	HK II Inhibitors	Cell Types	Reference
1	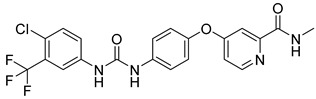	Inhibits growth factors and multiple kinases	3-BrPA 2-DG	Hepatocellular carcinoma (HCC); Hep3B Huh7	[97,98]
2	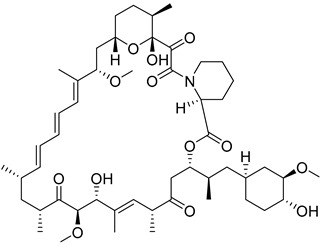	Inhibition of the mTOR pathway	3-BrPA	Human neuroblastoma (NB) cell NSCLC	[100,101]
3	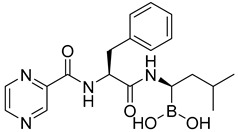	Inhibition of 26S proteasome chymotrypsin-like activity	3-BrPA	KMS-12-PE, KMS-11, H929, RPMI-8226, U266, MM.1S	[103]
4	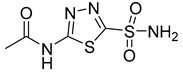	Inhibition of carbonic anhydrase and regulation of microenvironmental pH	3-BrPA	Huh-7	[108]
5	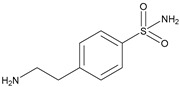	Inhibition of carbonic anhydrase and regulation of microenvironmental pH	3-BrPA	Huh-7, HepG2	[109]
6	NDV	Direct killing of cancer cells and activation of the immune system	D-mannoheptulose		[111]
7	M1 virus		2-DG		[112]
8	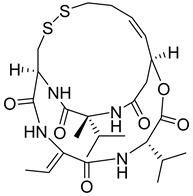	Inhibition of histone deacetylase activity	Clotrimazole, Bifoncarbazole	HCT-116, A549, 786-0, IGROV1, MDA-MB-231	[113]
9	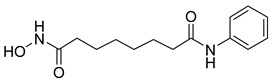	Inhibition of histone deacetylase activity	2-DG	H226	[114]
10	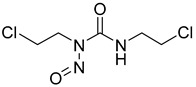	Cross-linking reactions occur with DNA to form DNA–DNA or DNA–protein crosslinks, which disrupt normal DNA replication and transcription	2-DG, 3-BP	SF763, SF126 SMMC-221, HepG2 SF763, SF126	[118,119,120]
11	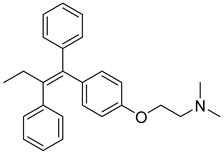	Competition with estrogen for estrogen receptors	As_2_O_3_	MCF7	[124]
12	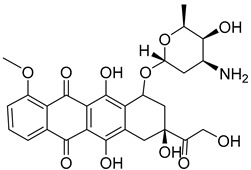	Inhibition of nucleic acid synthesis	Metformin		[127]
13	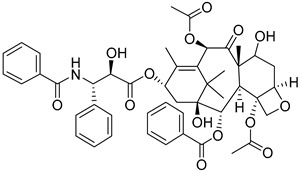	Enhancement of microtubule protein polymerization and inhibition of microtubule depolymerization that lead to the formation of stable, nonfunctional microtubule bundles and disruption of tumor cell mitosis	Metformin		[127]
14	Tralizumab	Inhibition of cancer cell growth by blocking the formation of HER2 receptor dimers	Metformin	NCI-N87, SNU216	[96,127]
15	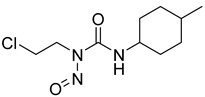	Destruction of DNA function	2-DG, 3-BrPA		[118,119,120]

**Table 2 molecules-29-00075-t002:** HK II inhibitors identified in this study.

Number	Name	Compound	Machine	IC_50_ for HK II	IC_50_ for Cells	References
1	G6P	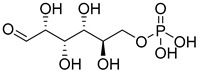		0.2 mM		[144]
2	T6P	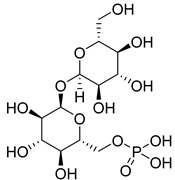	G6P analogs			[147]
3	Epigallocatechin gallate	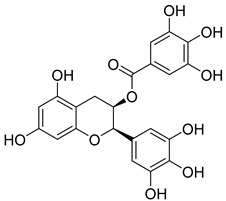	Bind HK Ⅱ pocket			[149,150]
4	Quercitrin	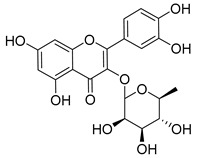				[150]
5	Dioscin	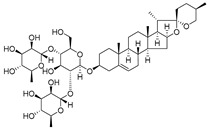	Targeted HK Ⅱ-VDAC1 binding		5 µM for HT29, HCT116, SW480	[151]
6	Matrine	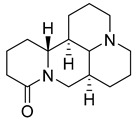			~0.5 mg/mL for K562, HL-60	[152]
7	Asiaticacid	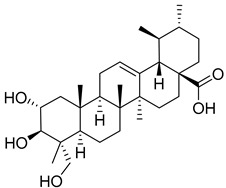				[153]
8	Andrographolide	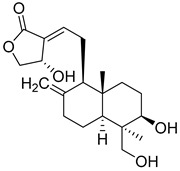				[153]
9	Bayogenin	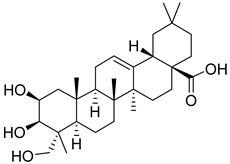				[153]
10	Pachymic Acid	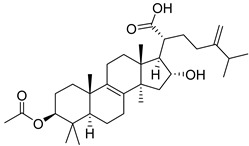	Promotion of HK II dissociation from mitochondria and cty-c (etc.)	5.01 µM	SK-BR-3 (5 µM)	[158]
11	α-Hederin	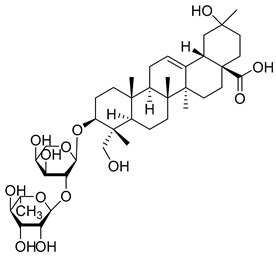			13.75 µM for NSCLC A549; 17.75 µM for NCI-H460; 18.04 µM for NCI-H292	[159]
12	Triptolide	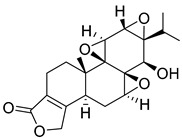				[163]
13	Wogonin	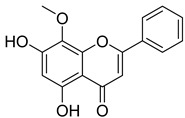				[164]
14	Curcumin	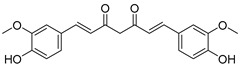	Induction of dissociation of HK II from mitochondria by phosphorylation of HK II via AKT			[169]
15	Methyl jasmonate	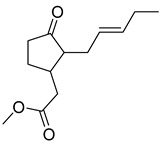		7.47 μM	4.17 mM for SKOV-3; 6.383 mM for A549	[167]
16	Methyl jasmonate derivative	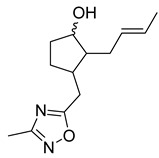		0.27 μM	1.772 mM for SKOV-3; 2.45 mM for A549	[167]
17	Sulforaphane	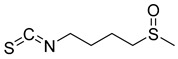			4.25 mM for A549, 1.772 mM for SK-OV3	[168]
18	Ketoconazole	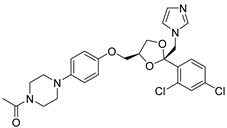				[115]
19	Posaconazole	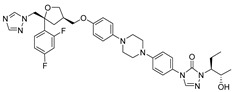				[115]
21	Glycyrrhetinic acid	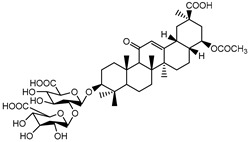			100 µg/mL	[160]
22	Steroids	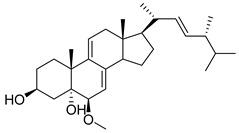		2.06 μM	5.05 µM for SW1990, 22.59 µM for Vero	[161]
23	Strepantibins A	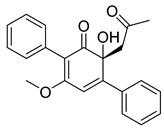		9.8 µM		[162]
24	Strepantibins B	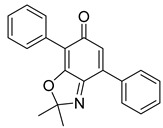		34.6 µM		[162]
25	Strepantibins C	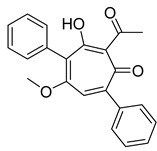		31.2 μM		[162]
26	Compound 27	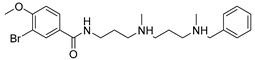		11.31 μM		[185]
27	11	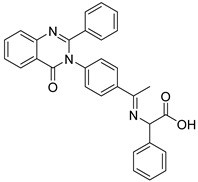			0.135 mM for MCF-7; 0.163 mM for MDA-231; 0.156 mM for PCL	[187]
28	20	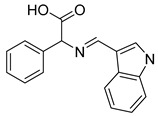			0.195 mM for MCF-7; 0.166 mM for MDA-231; 0.218 mM for PCL	[187]
29	Benserazide	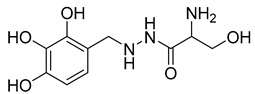		5.52 ± 0.17 mM		[186]

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
