# Peer review of "The Promoting Role of HK II in Tumor Development and the Research Progress of Its Inhibitors"

_molecules, 2023, doi:10.3390/molecules29010075_

Round 1

Reviewer 1 Report

Comments and Suggestions for Authors

In this paper, the authors presented a description of the structure of HK II and the specific biological factors that influence its role in cancer development, as well as the potential of HK II inhibitors in anti-tumor therapy. 

I have the following concerns (major):

The title is not accurate. HK II does not have antitumor activity, it's promoting tumor progression and development. 

Most figures are an oversimplification of the real picture. I suggest reducing the number of figures and making them self-explanatory.

Table 1 include the cell line or the cancer type

Table 2 is not well discussed in the text

Section drug repurposing is very poor. remove it or expand on it.

All chemical structures must be checked for accuracy. I detected some faulty structures.

Comments on the Quality of English Language

needs some editing 

Author Response

Dear Editor,

We have resubmitted our manuscript, “The Antitumor effects of Hexokinase II and its inhibitors” in Molecules. We thank you and the reviewer for the constructive comments. The manuscript is revised based on your comments. All changes are highlighted with red font. Here enclosed please also find a point-by-point response to the reviewer's comments.

Please do not hesitate to contact us with any further questions or recommendations.

Yours sincerely,

Yuji Wang

Department of Medicinal Chemistry, School of Pharmacy

Capital Medical University

[email protected]

Referee: 1 
Comments to the Author  

  1. The title is not accurate. HK II does not have antitumor activity, it's promoting tumor progression and development.

Response: Thanks for your comments. We changed the title to: Effect of hexokinase 2 and its inhibitors on tumor progression.

  1. Most figures are an oversimplification of the real picture. I suggest reducing the number of figures and making them self-explanatory.

Respouse: Thanks for your kind suggestion. We have added the explanation in the Figure legend. (Page 6, 12, 14 and 17)

  1. Table 1 include the cell line or the cancer type.

Response: Thanks for your kind comments. We will add the category of cell types to the manuscript. (page 34)

  1. Table 2 is not well discussed in the text. 

Respouse: Thanks for your kind comments. We will review and discuss Table 2 in more detail to ensure that the information is adequately presented and clearly explained and the text will also be revised. Thank you again for your advice. (page 25-27, line 670-744)

  1. Section drug repurposing is very poor, remove it or expand on it.

Respouse: Thanks for your kind comments. We will merge the contents with the above content to give it a more comprehensive presentation. (page 21, line 533-538; page 30, line 843-848)

  1. All chemical structures must be checked for accuracy. I detected some faulty structures.

Respouse: Thanks for your kind comments, We made revisions to the chemical structures involved in it. (page 38, Table 1, No.2、5、8、13)

Reviewer 2 Report

Comments and Suggestions for Authors

The review entitled “The Antitumor effects of Hexokinase II and its inhibitors” looks interesting in enlisting the approaches that have been adopted in inhibiting HKII thus far. The review however has certain shortcomings which need to be addressed for its publication in the journal Molecules. A detailed report of comments is attached separately.

Comments on the Quality of English Language

Quality of English language is overall reasonable.

Author Response

Dear Editor,

We have resubmitted our manuscript, “The Antitumor effects of Hexokinase II and its inhibitors” in Molecules. We thank you and the reviewer for the constructive comments. The manuscript is revised based on your comments. All changes are highlighted with red font. Here enclosed please also find a point-by-point response to the reviewer's comments.

Please do not hesitate to contact us with any further questions or recommendations.

Yours sincerely,

Yuji Wang

Department of Medicinal Chemistry, School of Pharmacy

Capital Medical University

[email protected]

Referee: 2 
The Antitumor effects of Hexokinase II and its inhibitors looks interesting

in enlisting the properties of HKII in great details, ranging from importance of its protein structure to the regulation of its promoter and RNA transcripts through various mechanisms. The review also talks about the approaches that have been adopted in inhibiting HKII in different tumors and sums up various pros and cons of different approaches employed thus far. The review, however, presents itself with various shortcomings which need to be addressed for its publication in the journal Molecules. A detailed report of comments is summarized below.

Line-wise comments

  1. The title of the review seems to contradict the subject matter since HKII has a well-known protumor activity and its inhibition has anti-tumor effects. The authors should revise the title.

Respouse: Thanks for your kind comment We changed the title to: Effect of hexokinase 2 and its inhibitors on tumor progression.

  1. Line 20-28: In addition to data references from China and the United States, please provide an account of the global incidences of different cancers and the role of hexokinases/metabolism in the reported incidence.

Respouse: Thanks for your kind comments. According to the estimates of cancer incidence and mortality compiled by the International Agency for Research on Cancer (IARC) in 2020, the latest information on the global burden of cancer is provided. It is predicted that there were approximately 19.3 million new cancer cases worldwide in 2020and approximately 10 million cancer deaths. And compared to 2020, the number will increase by 47% by 2040 [1].(page 29, line 823-828)

  1. Line 36: Use the word tissue before tumor or the word cells before normal. The statement is grammatically and logically inconsistent.

Respouse: Thanks for your kind comments. We checked and corrected the text of the paragraph, and thank you again for your valuable comments. (page 2, line 31-38)

  1. Line 66: Please change the term “research manuscript” to “review”.

Respouse: Thanks for your kind comments. The "research manuscript" has been changed to "review". “review”.

  1. The form is contextually inconsistent.

Respouse: Thank you for bringing that to my attention. I will review the form to ensure that it is consistent in its context.

  1. Lines 74-79: Please update the information by including the fifth hexokinase hexokinase domain containing 1 (HKDC1) to the list of hexokinases.

Respouse: Thanks for your kind comments, We will add this content to the document. (page 3, line 83)

  1. Figure 2: Please elaborate the figure legend.

Respouse: Thanks for your kind comments, We have supplemented the legend for Figure 2. (page 6)

Figure 2. HK II during glycolysis. ADP and PI in the cytoplasm are transported into the mitochondrial matrix through the mitochondrial inner membrane via the AAC protein. They participate in the TCA cycle. The ATP produced is transported from the mitochondrial matrix to the intermembrane space through the AAC protein and participates in the process of glucose conversion to G6P mediated by HK Ⅱ. The lactate produced is transported out of the cell through transport proteins on the cell membrane.AAC, Mitochondrial ADP/ATP carrier; M-state, matrix-open state, the AAC protein transports ADP into the mitochondrial matrix, with the structure of the AAC protein referred to as the m-state during this process; C-state, cytoplasmic-open state, the AAC protein transports ADP into the mitochondrial matrix, with the structure of the AAC protein referred to as the c-state during this process; TCA, tricarboxylic Acid Cycle;VDAC, voltage-dependent anion selective channel; HKⅡ,hexokinaseⅡ;G6P:Glucose 6-phosphate; F6P, fructose 6-phosphate; NDA+, Nicotinamide adenine dinucleotide; NDAH, Nicotinamide adenine dinucleotide(Prototype);

  1. Figure 3: Please elaborate the figure legend. Also, include a schematic and or a table to sum up all the information provided in the preceding section to make it reader friendly.

Respouse: Thanks for your kind comments, We have supplemented the legend for Figure 3. (page 12)

Figure 3. Proteins and signaling pathways that cause HKII up-regulation. ErbB2, tyrosine kinase receptor 2; HectH9, an enzyme belonging to the HECT (Homologous to the E6-AP Carboxyl Terminus) protein family; TRIM59, tripartite motif-containing protein 59; PTEN, phosphatase and tensin homolog; AKT, protein Kinase B; BACH1, BTB and CNC homology 1; Nrf2, nuclear factor erythroid 2-related factor 2; PIM2, pre-B lymphoma virus insert site 2; MAPK, mitogen-activated protein kinase; STAT3, transcription 3; Wnt/β-catenin, canonical Wnt/β-catenin pathway; C-myc, cellular-myelocytomatosis viral oncogene; Cyclin D1, G1/S-specific cell cycle protein-D1; CREB, cAMP-response element binding protein; AR, Androgen receptor; HIF-1α, hypoxia inducible factor 1 subunit alpha Gen; MZF1, myeloid zinc finger protein; SIRT6, a member of the histone deacetylase family; IL-1β, Interleukin-1β; , cicroRNA; ,miRNA; ,phosphorylation; ,ubiquitination; ,active;

  1. Figures 4 & 5: Please elaborate the figure legends and explain the differences in the mechanisms of processes outlined between normal and tumor cells. Also, include a schematic and/ or a table to sum up all the information provided in the preceding and succeeding sections. The presentation of information in sections 3-4 is overwhelming in the current form.

Respouse: Thanks for your kind comments, We have supplemented the legend for Figure 4 & 6(page 14 and 17).New figure 5 is a summary of Figure 3 and 4(page 14). The original Figure 5 has been changed to Figure 6(page 17).

Figure 4. Factors that contribute to the downward revision of HK II. Snial, snail family transcriptional repressor 1 Gene; CREB, cyclic-AMP response binding protein; KCNQ1OT1, a long non-coding RNA; IL13Rα1, interleukin-13 receptor subunit alpha-1.

Figure 5. Summary of various biological factors that impact HKⅡ expression

Figure 6. The mechanism by which HK II promotes drug resistance. After binding with VDAC, BCL-2 and/or HKII inhibit the release of cytochrome c from VDAC (left); When BAX and tBID are in the outer mitochondrial membrane, tBID binds to BAX/BAK, promoting their oligomerization and the release of cyt c (medium); HKⅡand BCL-xL reverse the translocation of tBID and BAX/BAK from the outer mitochondrial membrane to the cytoplasm (right).BCL-2, B-cell lymphoma-2;BAX/BAK,  BCL2-Associated X/K; Tbid, Bcl-2/BH3-only protein.

  1. Explain hierarchically about the altered metabolic regulation events which lead to altered/ enhanced utilization of HKII in tumor cells towards the advantage of tumor, explain different forms of tumor exhibiting this phenomenon and various challenges in their treatment, followed by current therapeutic avenues, further followed by targeting of HKII in isolation or in conjunction, benefits and limitations. Make the conclusions section more structured.

Respouse: Thanks for your kind comments, I will change to the following content:

Conclusion: Metabolic reprogramming plays a vital role in facilitating the growth, proliferation, and survival of cancer cells. Intrinsic mechanisms within these cells trigger the activation of signaling components that have a direct impact on enhancing the activity of metabolic enzymes or up-regulating transcription factors. Consequently, this leads to an increase in the expression of metabolic regulators [2, 3].

As a key rate-limiting enzyme in glycolysis, HK2 expression or overexpression is associated with poor prognosis, staging progression, metastasis and/or treatment resistance in various malignant tumors, including colorectal cancer[4], CC [5], lung cancer[6], breast cancer[7], HCC[8], prostate cancer[9] and other tumor types. The (over)expression of HK2 is also associated with maintaining the stemness of lung cancer cells and the survival rate of endometrial cancer in various tumor models[10-12]. Studies have shown that inhibiting or knocking out HK2 is beneficial for suppressing tumor development[13, 14].

Research has found that many proteins and transcription factors are associated with HK2, revealing complex mechanisms. Understanding the relationships between these proteins can enhance our understanding of HK2 regulation. By regulating and intervening in these proteins, we can further understand the role of HK2 in cellular metabolism and the mechanisms of related diseases.

However, when HK2 is inhibited, other energy-producing pathways may compensate for the decrease in glycolysis and maintain the survival of cancer cells[15]. Therefore, multitargeting can simultaneously target multiple metabolic pathways and produce a more pronounced inhibitory effect on cells, reducing the side effects of a single drug. In the previous discussion, we also mentioned examples of better inhibition when anticancer drugs are used in combination with HK2 inhibitors. However, drug combinations also face challenges, such as increased drug interactions and toxicity. Therefore, when designing multitargeting strategies, comprehensive evaluation and validation are needed. In conclusion, the strategy of multitargeting has potential advantages for the development of efficient anticancer drugs, but further research and practice are needed to verify their safety and effectiveness.

Since the glycolytic pathway is one of the essential processes for maintaining normal cell function, complete inhibition of glycolysis can lead to adverse reactions, limiting the value of this approach [15]. Therefore, selective inhibition of cancer-driven glycolysis is necessary to utilize this metabolic pathway for clinical cancer treatment. However, HK2 is still a promising target for anticancer treatment. (page 30-31, line 876-907)

Reference:

  1. Sung, H.; Ferlay, J.; Siegel, R. L.; Laversanne, M.; Soerjomataram, I.; Jemal, A.; Bray, F., Global Cancer Statistics 2020: GLOBOCAN Estimates of Incidence and Mortality Worldwide for 36 Cancers in 185 Countries. CA: A Cancer Journal for Clinicians 2021, 71, (3), 209-249.
  2. Perrin-Cocon, L.; Vidalain, P.-O.; Jacquemin, C.; Aublin-Gex, A.; Olmstead, K.; Panthu, B.; Rautureau, G. J. P.; André, P.; Nyczka, P.; Hütt, M.-T.; Amoedo, N.; Rossignol, R.; Filipp, F. V.; Lotteau, V.; Diaz, O., A hexokinase isoenzyme switch in human liver cancer cells promotes lipogenesis and enhances innate immunity. Communications Biology 2021, 4, (1), 217.
  3. Dey, P.; Kimmelman, A. C.; DePinho, R. A., Metabolic Codependencies in the Tumor Microenvironment. Cancer Discovery 2021, 11, (5), 1067-1081.
  4. Katagiri, M.; Karasawa, H.; Takagi, K.; Nakayama, S.; Yabuuchi, S.; Fujishima, F.; Naitoh, T.; Watanabe, M.; Suzuki, T.; Unno, M.; Sasano, H., Hexokinase 2 in colorectal cancer: a potent prognostic factor associated with glycolysis, proliferation and migration. Histol Histopathol. 2017, 32, (4), 351-360.
  5. Yang, H.; Hou, H.; Zhao, H.; Yu, T.; Hu, Y.; Hu, Y.; Guo, J., HK2 Is a Crucial Downstream Regulator of miR-148a for the Maintenance of Sphere-Forming Property and Cisplatin Resistance in Cervical Cancer Cells. Frontiers in Oncology 2021, 11.
  6. Yang, L.; Yan, X.; Chen, J.; Zhan, Q.; Hua, Y.; Xu, S.; Li, Z.; Wang, Z.; Dong, Y.; Zuo, D.; Xue, M.; Tang, Y.; Herschman, H. R.; Lu, S.; Shi, Q.; Wei, W., Hexokinase 2 discerns a novel circulating tumor cell population associated with poor prognosis in lung cancer patients. Proceedings of the National Academy of Sciences 2021, 118, (11), e2012228118.
  7. Lin, J.; Fang, W.; Xiang, Z.; Wang, Q.; Cheng, H.; Chen, S.; Fang, J.; Liu, J.; Wang, Q.; Lu, Z.; Ma, L., Glycolytic enzyme HK2 promotes PD-L1 expression and breast cancer cell immune evasion. Frontiers in Immunology 2023, 14.
  8. Li, M.; Jin, R.; Wang, W.; Zhang, T.; Sang, J.; Li, N.; Han, Q.; Zhao, W.; Li, C.; Liu, Z., STAT3 regulates glycolysis via targeting hexokinase 2 in hepatocellular carcinoma cells. Oncotarget; Vol 8, No 15 2017.
  9. Stein, M.; Lin, H.; Jeyamohan, C.; Dvorzhinski, D.; Gounder, M.; Bray, K.; Eddy, S.; Goodin, S.; White, E.; DiPaola, R. S., Targeting tumor metabolism with 2-deoxyglucose in patients with castrate-resistant prostate cancer and advanced malignancies. The Prostate 2010, 70, (13), 1388-1394.
  10. Wang, J.; Shao, F.; Yang, Y.; Wang, W.; Yang, X.; Li, R.; Cheng, H.; Sun, S.; Feng, X.; Gao, Y.; He, J.; Lu, Z., A non-metabolic function of hexokinase 2 in small cell lung cancer: promotes cancer cell stemness by increasing USP11-mediated CD133 stability. Cancer Communications 2022, 42, (10), 1008-1027.
  11. Dong, P.; Xiong, Y.; Konno, Y.; Ihira, K.; Kobayashi, N.; Yue, J.; Watari, H., Long non-coding RNA DLEU2 drives EMT and glycolysis in endometrial cancer through HK2 by competitively binding with miR-455 and by modulating the EZH2/miR-181a pathway. Journal of Experimental & Clinical Cancer Research 2021, 40, (1), 216.
  12. Li, H.; Song, J.; He, Y.; Liu, Y.; Liu, Z.; Sun, W.; Hu, W.; Lei, Q. Y.; Hu, X.; Chen, Z.; He, X. A.-O. X., CRISPR/Cas9 Screens Reveal that Hexokinase 2 Enhances Cancer Stemness and Tumorigenicity by Activating the ACSL4-Fatty Acid β-Oxidation Pathway. (2198-3844 (Electronic)).
  13. Patra, K. C.; Wang, Q.; Bhaskar, P. T.; Miller, L.; Wang, Z.; Wheaton, W.; Chandel, N.; Laakso, M.; Muller, W. J.; Allen, E. L.; Jha, A. K.; Smolen, G. A.; Clasquin, M. F.; Robey, B.; Hay, N., Hexokinase 2 is required for tumor initiation and maintenance and its systemic deletion is therapeutic in mouse models of cancer. (1878-3686 (Electronic)).
  14. Wu, Q.; Wang, S.-P.; Sun, X.-X.; Tao, Y.-F.; Yuan, X.-Q.; Chen, Q.-M.; Dai, L.; Li, C.-L.; Zhang, J.-Y.; Yang, A.-L., HuaChanSu suppresses tumor growth and interferes with glucose metabolism in hepatocellular carcinoma cells by restraining Hexokinase-2. The International Journal of Biochemistry & Cell Biology 2022, 142, 106123.
  15. Xu, S.; Catapang, A.; Braas, D.; Stiles, L.; Doh, H. M.; Lee, J. T.; Graeber, T. G.; Damoiseaux, R.; Shirihai, O.; Herschman, H. R., A precision therapeutic strategy for hexokinase 1-null, hexokinase 2-positive cancers. Cancer & Metabolism 2018, 6, (1), 7.

Round 2

Reviewer 1 Report

Comments and Suggestions for Authors

Thanks for addressing most of my comments. The review looks much better now. I still have a concern with the title. Please rephrase to reflect the true role of Hexokinase II as a tumour promotor and not suppressor. 

 Also, Table 2 needs to be fixed. Most of the structures are not complete and part of them is hidden behind the text.

Author Response

Dear Editor,

We have resubmitted our manuscript, “The promoting role of HK2 in tumor development and the research progress of its inhibitors” in Molecules. We thank you and the reviewer for the constructive comments. The manuscript is revised based on your comments. All changes are highlighted with red font. Here enclosed please also find a point-by-point response to the reviewer's comments.

Please do not hesitate to contact us with any further questions or recommendations.

Yours sincerely,

Yuji Wang

Department of Medicinal Chemistry, School of Pharmacy

Capital Medical University

[email protected]

Referee: 1 

Comments to the Author 

  1. Please rephrase to reflect the true role of Hexokinase II as a tumour promotor and not suppressor.

Response: Thank you for bringing this to my attention. The correct role of Hexokinase II is as a promoter of tumorigenesis, not as an inhibitor. I will revise my title to accurately reflect this as " Thank you for bringing this to my attention. The correct role of Hexokinase II is as a promoter of tumorigenesis, not as an inhibitor. I will revise my title to accurately reflect this as "The promoting role of HK2 in tumor development and the research progress of its inhibitors".".

  1. Also, Table 2 needs to be fixed. Most of the structures are not complete and part of them is hidden behind the text.

ResponseThanks for your kind comments. I will make the necessary adjustments to Table 2 to ensure that all structures are complete and visible. (page 31)"